# Photochemical Synthesis of Noble Metal Nanoparticles: Influence of Metal Salt Concentration on Size and Distribution

**DOI:** 10.3390/ijms241814018

**Published:** 2023-09-13

**Authors:** Shahad M. Aldebasi, Haja Tar, Abrar S. Alnafisah, Lotfi Beji, Noura Kouki, Fabrice Morlet-Savary, Fahad M. Alminderej, Lotfi M. Aroua, Jacques Lalevée

**Affiliations:** 1Department of Chemistry, College of Science, Qassim University, Buraidah 51452, Saudi Arabia; 411207279@qu.edu.sa (S.M.A.); alnafisaha@qu.edu.sa (A.S.A.); n.kouki@qu.edu.sa (N.K.); f.alminderej@qu.edu.sa (F.M.A.); lm.aroua@qu.edu.sa (L.M.A.); 2Department of Physics, College of Sciences and Arts at ArRass, Qassim University, Buraidah 51452, Saudi Arabia; l.beji@qu.edu.sa; 3CNRS, IS2M UMR 7361, Université de Haute-Alsace, F-68100 Mulhouse, France; fabrice.morlet-savary@uha.fr (F.M.-S.); jacques.lalevee@uha.fr (J.L.)

**Keywords:** noble metals, gold nanoparticles, silver nanoparticles, benzophenone, photoinitiator, photoreduction, photosynthesis

## Abstract

This paper explores the photochemical synthesis of noble metal nanoparticles, specifically gold (Au) and silver (Ag) nanoparticles, using a one-component photoinitiator system. The synthesis process involves visible light irradiation at a wavelength of 419 nm and an intensity of 250 mW/cm^2^. The radical-generating capabilities of the photoinitiators were evaluated using electron spin resonance (ESR) spectroscopy. The main objective of this study was to investigate how the concentration of metal salts influences the size and distribution of the nanoparticles. Proposed mechanisms for the photochemical formation of nanoparticles through photoinitiated radicals were validated using cyclic voltammetry. The results showed that the concentration of AgNO_3_ significantly impacted the size of silver nanoparticles, with diameters ranging from 1 to 5 nm at 1 wt% and 3 wt% concentrations, while increasing the concentration to 5 wt% led to an increase in the diameter of silver nanoparticles to 16 nm. When HAuCl_4_ was used instead of AgNO_3_, it was found that the average diameters of gold nanoparticles synthesized using both photoinitiators at different concentrations ranged between 1 and 4 nm. The findings suggest that variations in HAuCl_4_ concentration have minimal impact on the size of gold nanoparticles. The photoproduction of AuNPs was shown to be thermodynamically favorable, with the reduction of HAuCl_4_ to Au^0^ having ∆G values of approximately −3.51 and −2.96 eV for photoinitiators A and B, respectively. Furthermore, the photoreduction of Ag^+1^ to Ag^0^ was demonstrated to be thermodynamically feasible, with ∆G values of approximately −3.459 and −2.91 eV for photoinitiators A and B, respectively, confirming the effectiveness of the new photoinitiators on the production of nanoparticles. The synthesis of nanoparticles was monitored using UV-vis absorption spectroscopy, and their sizes were determined through particle size analysis of transmission electron microscopy (TEM) images.

## 1. Introduction

Colloidal silver nanoparticles (AgNPs) and gold nanoparticles (AuNPs) have garnered significant attention in recent years due to their utilization in various fields, such as materials science, biotechnology, and organic chemistry, for their ability to act as molecular markers in diagnostic imaging, catalysis, and other applications [1,2,3,4,5]. These noble metals exhibit unique absorption properties and optical characteristics when their size is less than 100 nm. Metal nanoparticles possess more distinct and intriguing properties than atoms, surfaces, or macromolecular materials [6,7,8]. Because of their strong light scattering capabilities, AgNPs and AuNPs can be used in optical, imaging, and sensing applications. Given the advantages that nanostructured metals have over their molecular counterparts, it is logical to make use of the benefits of metal colloids produced through different synthesis methods [9,10].

Au and Ag NPs are widely used due to their stability and superior quality factor of localized surface plasmon resonance (LSPR) compared to other metal NPs [11,12]. The prospect of tuning the optical properties of NPs makes them versatile for use in various applications [13,14]. This can be achieved by changing the size, shape, or composition of the NPs. Changing the size and shape allows the LSPR to be positioned in a narrow spectral range, while changing the composition allows the LSPR to be positioned throughout a broad spectral range. For example, spherical Ag and Au NPs of various sizes cover the 400–450 nm and 520–570 nm spectral ranges [15], respectively. Various synthesis methods, including physical, chemical, and photochemical, have been developed to produce metal NPs with various sizes, shapes, compositions, and structures, with photochemical synthesis being a particularly preferred method [16,17,18,19,20,21]. Photochemical synthesis offers several advantages, including fewer byproducts, clean synthesis, strict irradiation control, simple and inexpensive equipment, a lower temperature, a lower overall energy requirement to drive the reaction, and spatiotemporal control over the rate and degree of reduction [22,23,24,25,26,27,28]. This method mainly depends on photoinitiators, which have a major impact on the NPs properties.

The development of new, efficient, and safer photoinitiators is a promising avenue for advancing the field of nanoparticle synthesis. The photoreduction of metallic salts to metallic nanoparticles is primarily caused by the radicals produced when light interacts with the chemical constituents of the solution. In addition to the radicals produced by the photoinitiator, the type of photoinitiator plays a significant role in determining the nanoparticles’ final size, shape, and distribution. Photoinitiators have been classified into two types based on their optical behavior: (i) Type I, also known as α-cleavage, which provides the scope for initiating radicals via bond cleavage processes upon light absorption. (ii) Type II, which undergoes photoexcitation followed by electron or hydrogen transfer processes and, as a result, the formation of initiating species [29,30,31].

One of the most common types of photoinitiators, benzoin and its derivatives are capable of unimolecular bond cleavage upon light irradiation [29]. Ketone components are critically important as Type II photoinitiators. For instance, benzophenone [32], thioxanthone [33], camphorquinone [34], etc. all show bimolecular photo-behavior in the formation of initiating species. The photoexcitation of the photoinitiator and its excited state interaction with other components, called co-initiators, through various transfer processes are crucial to the observable behavior. When compared to Type I photoinitiators, which require high-energy, short-wavelength light to undergo bond cleavage, Type II photoinitiators are preferable due to their lower energy requirements. Type II photoinitiators, on the other hand, exhibit absorption characteristics at higher wavelengths and can be designed and decorated to manipulate their optical behavior, thereby increasing their spectral sensitivity to the visible ranges of the electromagnetic spectrum. It is cost-effective because it makes use of low-energy lighting sources [35]. Photoinitiators have many uses beyond nanoparticle production. Photoinitiators have also found applications in the biomedical field, where they are used in various approaches such as photodynamic therapy and drug delivery systems [36,37]. The process of polymerization in 3D printing is initiated by photoinitiators [38]. Photoinitiators also polymerize monomers on surfaces in photo-induced graft polymerization [39].

Several researchers have used Type I and Type II initiators to photochemically produce Ag, Au, and Cu nanoparticles [26,28,40,41,42,43]. The excellent light absorption characteristics of benzophenone (BP) and its derivatives make them one of the most widely used Type II photoinitiators in many technologically significant UV-curing applications [44]. Benzophenones are excited to the singlet excited state when exposed to UV light. Through intersystem crossing, the singlet excited state decays into the triplet excited state. The ketone derivatives in the triplet excited state absorb hydrogen from the hydrogen donor such as alcohols [45], amines [32], ethers [46], or thiols [47], leading to the formation of a radical produced from the carbonyl compound (ketyl-type radical) and another radical derived from the hydrogen donor [45]. These radicals reduce metal ions to produce metal nanoparticles [48].

The ketyl radicals formed during the photoreduction of ketones are excellent reducing agents, effectively converting Ag^+^ ions to Ag^0^. The photoreduction of benzophenone is a reliable source of ketyl radicals; in the presence of Ag^+^, the mechanism is thought to involve electron transfer by ketyl radicals. Kometani et al. [49] found that when an aqueous solution of AgClO_4_, sodium dodecyl sulfate, and benzophenone was irradiated with near-UV light, photoreduction of Ag^+^ ions occurred, resulting in the formation of colloidal silver nanoparticles. Eustis et al. [50] investigated the process of producing silver nanoparticles from benzophenone using both a laser and a mercury lamp as light sources. These researchers also modified silica with benzophenone, which helped to create stable silver nanoparticles using a solid-supported photosensitizer [51]. Scaiano et al. [52] also prepared micelles of sodium dodecyl sulfate and used 1,4-cyclohexadiene as a hydrogen donor to promote the rapid generation of ketyl radicals. Irradiating this system in the presence of Ag^+^ and BP leads to the rapid and efficient formation of silver nanoparticles. In a subsequent study, Sakamoto et al. identified the fabrication of gold nanoparticles in a poly(vinyl alcohol) film using a two-color, two-laser irradiation with benzophenone as the reducing agent [53]. Another study showed that AuNPs can be produced photochemically under the irradiation of a 368 nm LED, exploiting the rarely reported nucleophilic property of the benzophenone triplet [54]. Sakamoto et al. also found that co-reducing Au and Cu ions in a PVA film resulted in the creation of core/shell-like Au/Cu bimetallic NPs [55]. In their study, the PVA film containing BP, Cu(acac)_2_, and HAuCl_4_ was irradiated with UV light. Zhao et al. [56] explored the mechanism of BP-initiated one-step photosynthesis of silver nanoparticles in a system free of hydrogen donors. This investigation was conducted through a combination of spectroscopic and theoretical methods. The electron donation ability of BP triplets leads to the direct reduction of Ag^+^ to Ag^0^. In a recent study, a combination of triethylamine and iodonium salt, which are benzophenone derivatives, were tested as potential new photoinitiators for the rapid and efficient formation of metal nanoparticles in an organic solvent, where silver and gold ions were reduced under light at 419 nm [32].

Thioxanthone (TX) and its derivatives are widely used as Type II photoinitiators [57]. Thioxanthone-Anthracene (TX-A) was recently used as a one-component photoreducing agent to produce metal and metal oxide nanoparticles from metallic salts like AgNO_3_, HAuCI_4_, and MnCI_2_ [58]. The AuNPs (18–25 nm) synthesized in the polymer matrix are larger than the AuNPs (8–16 nm) synthesized in the solution with an air atmosphere. It is crucial to mention how the atmosphere affects the size and shape of the NPs. In solution, AgNPs in the size range of 45–53 nm can form due to endoperoxide formation in the air atmosphere, whereas in the nitrogen atmosphere, AgNPs can grow as large as 1900 nm. Arsu and co-workers also investigated the effect of a photoreducing agent on the size, shape, and distribution of metallic Ag and Au nanoparticles in the presence of a one-component Type II photoinitiator, namely 2-thioxanthone thioacetic acid dioxide [59]. The rapid achievement of in situ formation of metallic nanoparticles was observed in both aqueous solution and polymer matrix. The impact of the photoinitiator concentration on the size, shape, and distribution of the nanoparticles was examined. The diameters of the AuNPs synthesized with a photoinitiator concentration of 2 × 10^−3^ M are approximately 90 nm. However, the nanoparticle size increases to 1.623 μm upon decreasing the photoinitiator concentration. On the other hand, the observed size of AgNPs is 143.4 nm at a low photoinitiator concentration, whereas the size of AgNPs increases to 427.2 nm upon an increase in photoinitiator concentration.

Our paper presents the development of novel photoinitiators specifically designed for the efficient synthesis of silver and gold nanoparticles using photochemical methods. Our main focus is to enhance the effectiveness and safety of Ag and Au NPs production. We systematically investigate the impact of different concentrations of Au and Ag precursors on the size and morphology of the resulting NPs. Through rigorous analysis, we elucidate the significant role of precursor concentrations in controlling the size and morphology of the NPs, and propose a comprehensive mechanistic understanding of the underlying reaction. This research contributes to the advancement of NP synthesis techniques and provides valuable insights into the key factors influencing their formation.

## 2. Results and Discussion

### 2.1. Light Absorption Properties of the Photoinitiators

The UV-vis absorption spectra of photoinitiators A and B, each at a concentration of 5 × 10^−4^ M, in methanol are shown in Figure 1. Photoinitiator A exhibits maximum absorption peaks at 270 nm (π-π* transition) and 324 nm (n-π* transition) in methanol, with a bathochromic shift of 20 nm and hyposochromic shift of 6 nm, compared to the reference compound BP (250 nm and 350 nm, respectively) [60]. It is known that transitions of BP in the region of 250 nm are of the π-π* type. The important n-π* transitions are typically found between 300 and 350 nm and are characterized by a low extinction coefficient due to the spin forbidden transition [61]. In contrast, photoinitiator B exhibits a maximum absorption wavelength at 334 nm, which is likely the n-π* transition.

### 2.2. Photolysis of Photoinitiators

To study the photolysis of the photoinitiators, the evolution of the UV-vis spectra of their solutions in methanol is monitored under light with a wavelength of 419 nm and different exposure times. Figure 2 and Figure 3 show the results of this experiment. When the photoinitiators are irradiated in methanol using a photoreactor consisting of ten LED lamps at 419 nm with an irradiation intensity of 250 mW/cm^2^, it is observed that the ground-state absorption decreased, and there were no colored photolysis products as the UV light exposure time increased for both photoinitiators. This indicates that these photoinitiators are not photochemically stable. In fact, the photolysis of photoinitiator B is faster than that of photoinitiator A, which requires more time. Figure 2a shows a plot of UV absorption versus wavelength and demonstrates the decrease in absorbance. When the irradiation time was extended to 116 min, a decrease in absorbance at 272 and 330 nm was observed, and it almost disappeared. In contrast, Figure 3a shows that the maximum absorbance at 330 nm almost disappeared within 98 min. Figure 2b and Figure 3b show that the peaks of the absorbance follow an exponential decay. In the case of photoinitiator A, the decay time τ is around 4406 s and 5968 s for the peaks at 330 nm and 272 nm, respectively. In the case of photoinitiator B, the decay time τ is about 2512 s for the only observed peak at 330 nm, which is much lower than that obtained using photoinitiator A. We can conclude that the absorbance extinction is faster in the case of photoinitiator B.

According to previous findings for other carboxylic acid derivatives of benzophenone, the proposed mechanism for the initiation of photoinitiator A is based on intermolecular hydrogen abstraction and decarboxylation (see Figure 1). The involvement of a decarboxylation process during the photolysis of A was confirmed by detecting the presence of CO_2_ using a procedure described in the literature [62]. A 1 mL solution of photoinitiator A in DMF (5 mM) was placed in a Pyrex tube, which was connected to another tube containing an aqueous solution of Na_2_CO_3_ (0.67 mM) and a drop of phenolphthalein solution (0.025 mM). After 3 h of irradiation, the pink color of the phenolphthalein solution disappeared, indicating the formation of CO_2_.

It has been demonstrated that the excited state of a photoinitiator can abstract hydrogen from the ground state of another molecule, resulting in the generation of an alkyl radical through photodecarboxylation. Additionally, it has been shown that self-quenching from the triplet state can lead to the formation of initiating radicals during the decarboxylation process for this type of benzophenone derivative.

The ability of both photoinitiators A and B to generate radical species under light irradiation was analyzed through ESR spectroscopy. In Figure 4, the acquired ESR spectra after 100 s of light irradiation at 405 nm are shown. The simulation of both spectra led to the following hyperfine coupling constants, (hfc) a_N_ = 13.5 G and a_H_ = 1.8 G, which are characteristic of oxygen-centered radicals, typically the 0 °C ones. The production of free radicals for photoinitiators A and B upon light irradiation was evident, but the chemical mechanism for photoinitiator B remains unclear in terms of the yield and the nature of the radical center.

### 2.3. Photoinitiators Oxidation Process 

The oxidation potentials (E_ox_) of photoinitiators A and B were determined using cyclic voltammetry (see Figure 5). The excited state energies ($E^{*}$) were obtained from the intersection of the absorption and luminescence spectra (see Appendix A). These values are presented in Table 1. The initial step in this work is to understand the thermodynamic formation of nanoparticles using photoinitiators as reducing agents. According to Newman et al. [63], the reduction potential of HAuCl_4_ to Au^0^ was found to be 0.854 V. Moreover, the reduction potential of Ag^+^/Ag was found to be 0.799 V by Kornweitz et al. [10]. The free energy change (∆G) for the photoproduction of NPs was evaluated using Equation (1). We find that the photoproduction of AuNPs is shown to be thermodynamically favorable. The reduction of HAuCl_4_ to Au^0^ had a ∆G value of approximately −3.514 and −2.964 eV for photoinitiators A and B, respectively, indicating that the process was favorable (as indicated in Table 1). When exposed to UV light, photoinitiators are excited to the singlet excited state. This state then decays into the triplet excited state through intersystem crossing. In the triplet excited state, the benzophenone undergoes hydrogen abstraction to generate the ketyl radical, which reduces Au^+3^ to Au^+2^. Au^+2^ is unstable and can be reduced further by the radical to Au^+1^ and then to Au^0^, leading to the formation of nanoparticles [44]. On the other hand, the photoreduction of Ag^+1^ to Ag^0^ was demonstrated to be thermodynamically feasible, with ∆G values of approximately −3.459 and −2.909 eV for photoinitiators A and B, respectively (Table 1). Ketyl radicals, produced in the photoreduction of ketones such as benzophenone for photoinitiator A, are highly potent reducing agents and can effectively convert Ag^+^ to Ag^0^ [52].

### 2.4. Photoproduction of Metal NPs

The new photoinitiators A and B do not promote a thermal reduction of gold III and silver ion. It was tested under 50 and 80 °C in dark conditions using the weight percentages 1, 3, and 5 wt%. These solutions were stirred for a long time (around 50 min), and it does not affect the color of the samples or create a SPR peak. The last one indicates that the thermal reduction does not occur. For this reason, it was interesting to study the photochemical reduction of gold III and silver ion by the new photoinitiators A and B.

#### 2.4.1. Photoproduction of Au Nanoparticles

Gold nanoparticles were synthesized via photoreduction, which was repeated using photoinitiators A and B. HAuCl_4_ solutions at three different concentrations (1, 3, and 5 wt%) with the photoinitiators were exposed to a 419 nm LED at 250 mW/cm^2^ for a certain period of time to study the effect of Au^+3^ concentration on the production of Au^0^. Upon irradiation, the transparent solutions turned purple, indicating AuNP production. AuNP formation was monitored with UV-vis spectroscopy. The obtained solution, when left at room temperature for 7 days, maintains its transparency without precipitating. These findings suggest that photoinitiators A and B behave as reductants and stabilizers and selectively produce AuNP photoirradiation.

For the AuA1 (1 wt%) solution in the presence of photoinitiator A, AuNP synthesis took approximately 40 min. Surface plasmon resonance (SPR) appeared after 60 s of irradiation at 541–579 nm, corresponding to AuNPs averaging 3.45 ± 0.20 nm. Increasing absorbance over time showed ongoing AuNP growth (see Appendix A).

By increasing the concentration of HAuCl_4_ to 3 wt% (AuA2), we noticed that the SPR remained constant while increasing the absorbance intensity peak versus irradiation time during AuNPs synthesis. The SPR was obtained at a wavelength of 550 nm and with particle sizes of about (3.38 ± 0.20) nm, as shown in Appendix A.

In the AuA3 solution (5 wt%), we observed that the wavelength had an oscillatory shift, and an increasing absorbance intensity peak versus irradiation time was also observed. The SPR is obtained at the wavelength range 553–567 nm, and AuNPs have a mean diameter of about (2.53 ± 0.20) nm (see Figure 6).

During this study, we noticed that the concentration of (HAuCl_4_) was found to affect the absorption of the AuNPs, as the absorption maximum (λ_max_) showed a hypsochromic shift when the concentration of HAuCl_4_ was varied from 1 wt% to 5 wt%. The proposed reaction mechanism is illustrated in Figure 2(1).

The synthesis of gold nanoparticles was also studied using photoinitiator B with varying concentrations of HAuCl_4_. For 1 wt% (AuB1), a wavelength blue shift and an increasing absorbance intensity peak versus irradiation time during AuNPs synthesis were noticed. The surface plasmon resonance (SPR) manifested after 360 s of irradiation time at a wavelength range of 533–541 nm, and the obtained AuNPs have a mean diameter of about (3.15 ± 0.20) nm (see Appendix A). For the concentration of 3 wt% of gold chloride added to the photoinitiator B (AuB2), it was observed that the wavelength remained constant while increasing the absorbance intensity peak versus irradiation time during AuNPs synthesis. By contrast, surface plasmon resonance (SPR) was observed after 60 s of irradiation time at a wavelength of 539 nm, as depicted in Appendix A. The obtained AuNPs have a mean diameter of about (3.80 ± 0.20) nm, which is comparable with the 1 wt% of HAuCl_4_. Figure 7 shows the photoreduction of gold nanoparticles using 5 wt% of HAuCl_4_ (AuB3). It was noticed that by using this concentration the surface plasmon resonance (SPR) manifested after 240 s of irradiation time at a wavelength range of 546–550 nm, and the obtained AuNPs have a mean diameter of about (2.75 ± 0.20) nm, which is comparatively smaller than the previous concentrations. The results show that the concentration of HAuCl_4_ had an effect on the absorption of the AuNPs: the absorption increased as the concentration increased.

In order to understand the photoreduction mechanism of gold(III) by photoinitiator B for the synthesis of AuNPs, we carried out an investigation into the effect of various pH values in both acidic and basic environments (see Section 3.8). Our results suggest that the formation of AuNPs in the photoreduction process with photoinitiator B follows a nucleation/growth mechanism [64] (see Figure 3). The coordination of photoinitiator B with Au^+3^Cl_4_^−^ leads to the formation of a photoinitiator B–Au^+3^Cl_2_^−^ complex (Figure 3(1)), which was confirmed with the absorption spectra. The absorption spectra show that methanol containing 3 wt% gold(III) chloride exhibits a ligand-to-metal charge transfer (LMCT) band at 325 nm (green line in Figure 8a). After 16 min of irradiation, the appearance of a band at 298 nm (purple line) confirmed the formation of the photoinitiator B–Au^+3^Cl_2_ complex. Irradiation of the photoinitiator B–Au^+3^Cl_2_ complex results in a decrease in the LMCT band and the appearance of the SPR band of AuNPs. The photoinitiator B anion can be adsorbed onto the surface of Au^0^ (Figure 3(2)), acting as a surface stabilizer [65]. The negative charge of the anions suppresses the aggregation of AuNPs due to electrostatic repulsion [65,66]. As illustrated in Figure 8b, the SPR band of the solution increased at pH 6 and 12. However, at pH 2, photoinitiator B retains its molecular form, indicating that it has not undergone deprotonation. Consequently, a complex is not formed between the compound and gold(III).

The Au^+3^ absorbance calculations revealed that when using photoinitiator A, the rate of conversion of gold III to gold nanoparticles increases as the concentration is increased. At 1 wt%, the conversion rate is 31%, whereas at 5 wt%, the conversion rate increases to 71%. It was noted that the rate of conversion of Au^+3^ to Au^0^ using photoinitiator B did not increase with increasing concentration. Instead, it remained constant within the range of 74% (see Table 2).

#### 2.4.2. Photoproduction of Ag Nanoparticles

Using different concentrations of silver salt (AgNO_3_) as a precursor in methanol and photoinitiators as reducing agents, silver ions were photoreduced to silver nanoparticles (AgNPs) in this experiment (see Section 3.7). For a certain period of time, the solution was exposed to an LED with a wavelength of 419 nm and an intensity of 250 (mW/cm^2^). UV-vis absorption spectroscopy was used to keep track of the synthesis of AgNPs. After being exposed to radiation, the solution changed from being clear to orange, showing that silver ions had been successfully reduced to AgNPs. This investigation was conducted with the two photoinitiators to create silver nanoparticles. The effect of silver nitrate concentration on the production of AgNPs was evaluated using three different concentrations of silver nitrate (1, 3, and 5 wt%).

The growth of AgNPs using the ketyl radical of benzophenone derivate (photoinitiator A) was efficiently achieved within approximately 4 min. The first significant SPR absorption of the AgNPs at different concentrations occurs after 2 min of irradiation time at 422–430 nm. This was followed by sublinear growth of the surface plasmon absorption as a function of time (Figure 9). Moreover, the absorption spectra show increased absorbance intensity peaks during the synthesis of AgNPs, which indicates an increase in the concentration of AgNPs [67].

At low concentrations, the appearance of surface plasmon resonance (SPR) was observed following an irradiation time of 120 s, within the wavelength range of 419–438 nm (see Appendix A). The AgNPs acquired exhibit an average diameter of approximately (2.30 ± 0.20) nm. When the concentration is increased to 3 wt%, we notice that the SPR peak appears at 422–430 nm after 60 s of irradiation time, and the obtained AgNPs possess an average diameter of approximately (2.90 ± 0.20) nm. It is observed that the nanoparticle sizes of 1 wt% and 3 wt% were similar; however, at elevated concentrations of 5 wt%, there was a discernible increase in size due to the formation of aggregates. As a result, the observation SPR was at 435–426 nm and the average size of the collected AgNPs was (16 ± 0.20) nm, as shown in Figure 9. The proposed reaction mechanism is illustrated in Figure 2(2).

The synthesis of silver nanoparticles (AgNPs) using photoinitiator **B** at different concentrations was observed to exhibit a characteristic surface plasmon resonance (SPR) at wavelengths ranging from 419 to 435 nm, which increased with irradiation time from 0 to 4 min, indicating continued generation of AgNPs (Figure 10). The concentration of silver nitrate (AgNO_3_) was found to affect the absorption of the AgNPs versus irradiation time, as shown in Appendix A. The SPR’s absorption spectra revealed a relationship between the redshift wavelength and the elevated peak in absorbance intensity during the production of AgNPs. Notably, a faster onset of SPR growth was observed in samples with lower AgNO_3_ concentrations (1 wt%) compared to higher concentrations (3 wt% and 5 wt%). TEM analysis of the nanoparticles further revealed an average size range of 1–18 nm. Specifically, at 1 wt% concentration, the appearance of SPR was observed after 60 s of irradiation time at a wavelength range of 409–430 nm with an average diameter of (2.00 ± 0.20) nm. Increasing the concentration to 3 wt% showed an SPR band after 120 s of irradiation time at a wavelength range of 420–428 nm, and the obtained AgNPs had a mean diameter of (5.10 ± 0.20) nm, as confirmed with TEM images in Appendix A (AgB1 and AgB2). At high concentrations, a redshift was observed at a wavelength of 433 nm, with an increase in nanoparticle size to (14.00 ± 0.20) nm, as depicted in Figure 10 (AgB3). Furthermore, photoinitiator B exhibited aggregations at high concentrations, similar to the behavior observed with photoinitiator A.

As we discussed earlier, both acidic and basic media pH (2, 6, and 12) were tested to see how pH affects the photoreduction process of silver nitrate by photoinitiator B during AgNPs synthesis (see Section 3.8). Figure 11a demonstrates a similar result to that observed with AuNPs, where there is a decrease in the LMCT band and the appearance of the SPR band of AgNPs at 340 nm. Moreover, the solution’s SPR band exhibited an increase at pH levels of 6 and 12, as depicted in Figure 11b. At a pH of 2, it can be observed that photoinitiator B remains in its molecular form, suggesting that deprotonation has not occurred. As a result, the formation of a complex between the compound and silver ion does not occur.

When comparing the growth of AgNPs obtained using the two photoinitiators under the same conditions, it was observed that both photoinitiators offer rapid particle generation. Furthermore, the results indicate that the average particle size of silver nanoparticles exhibited an increase with increasing concentrations of silver nitrate for both photoinitiators, as determined with the measurement (see Figure 12a). Similar observations were reported by Goh et al. [68] and Bicer and Sisman [69] in their previous works, indicating that reducing the concentration of metal salt can result in the production of smaller particles. According to theoretical considerations, the rate of conversion of ions to metal particles in a reaction is dependent on the initial concentration of metal ions [70]. For Figure 12b concerning AuNPs, the reduction in size may have its origin in the stripping of Au aggregates by chlorine after a certain critical concentration of HAuCl_4_. Choosing a different Au precursor is preferable. A high SPR value is observed when compared to the nanoparticle size. It is assumed that the observed phenomenon could be attributed to the influence of the solvent used [71].

## 3. Materials and Methods

### 3.1. Materials

The structures of the photoinitiators are shown in Figure 4, and the synthetic process for photoinitiator B can be found in the Appendix A [72]. Moreover, Appendix A illustrate the ^1^H NMR and ^13^C NMR spectra of photoinitiator B, respectively. Photoinitiator A was synthesized in a previous study [73]. Silver nitrate (AgNO_3_, 99.99%), gold (III) chloride hydrate (HAuCl_4_, 99.99%) and methanol are purchased from Sigma Aldrich (Burlington, MA, USA).

### 3.2. Irradiation Source

The prepared solution was placed in a pyrex tube having a quartz window ((i.d.) 9 mm), and irradiated via an LED lamp at 419 nm with an intensity of 250 mW/cm^2^ in standard conditions.

### 3.3. Absorbance Measurements

The absorption properties of the photoinitiators evolution was followed using a Shimadzu UV-1800 spectrophotometer (Shimadzu, Duisburg, Germany).

### 3.4. ESR Experiments

The ESR spectra were recorded at room temperature using an X band spectrometer (EMXPlus, Bruker, Germany, Karlsruhe). Chemicals are dissolved in tert-butyl benzene, poured in a quartz EPR tube and adding N-tert-butyl-alpha-phenylnitrone (PBN) as a spin trap agent, and then, nitrogen was saturated via subsequent gas bubbling. Samples are irradiated inside the EPR cavity using LED emission at 405 nm (Thorlabs M405LP1). Spectra are simulated using the Winsim v.0.96 software.

### 3.5. Redox Potentials

The oxidation potential (*E_ox_*) in acetonitrile solution for the photoinitiators A and B were estimated using cyclic voltammetry with tetrabutylammonium hexafluorophosphate 0.1 M (TBAP) as a supporting electrolyte. The potential of the working electrode was gauged against the Ag/AgCl reference electrode (E^°^ = 0.203 V versus standard hydrogen electrode (SHE)), a pure Pt wire was utilized as the counter electrode, and a platinum rod with a 0.2 cm^2^ surface area was utilized as the working electrode. The free energy change $\Delta G_{et}$ for an electron transfer reaction is calculated using the classic Rehm–Weller equation (Equation (1)).
(1)$$\Delta G_{et}=E_{ox}-E_{red}-E^{*}+C$$
where $E_{ox}$, $E_{red}$, $E^{*}$, and C are the oxidation potential of the electron donor, the reduction potential of the electron acceptor, the excited state energy level, and the Coulombic term for the initially formed ion pair, respectively [74]. It often occurs that C is neglected in polar solvents, which is the case here.

### 3.6. Fluorescence Experiments

A JASCO FP-8200 spectrometer (JASCO, Riyadh, Saudi Arabia) was used to determine the fluorescence properties of the photoinitiators A and B in methanol, each at a concentration of 1 × 10^−4^ M.

### 3.7. Photoproduction of Gold/Silver Nanoparticles by Photoinitiators in Methanol Solution

The gold chloride and silver nitrate were photoreduced to nanoparticle sizes using two photoinitiator systems in a methanol solution. Both gold chloride and silver nitrate were dissolved in methanol in all samples. This work was studied with varying concentrations of HAuCl_4_ and AgNO_3_, as indicated in Table 3. The concentration of both photoinitiators solutions was fixed at (1 × 10^−4^ M) for all samples. The solution contained 1 mL of the metal salt solution and 2 mL of the photoinitiator. The solution was exposed to an LED with a wavelength of 419 nm and an intensity of 250 (mW/cm^2^) for a certain period of time. The evolution of the SPR nanoparticles was continuously followed using a Shimadzu UV-1800 spectrophotometer (Shimadzu, Duisburg, Germany).

### 3.8. Photoproduction of Gold/Silver Nanoparticles by Photoinitiators B at Different pH Values

To understand the photoreduction mechanism of gold(III) by photoinitiator **B** for the production of AuNPs, the effects of different pH values were investigated in both acidic and basic media. The buffer solutions were prepared at different pH values: 2, 6 and 12. The samples were prepared by combining 1 mL of 3 wt% gold(III) with 2 mL of photoinitiator B (1 × 10^−4^ M) in methanol. The reaction was carried out at 25 °C under irradiation with a wavelength of 419 nm and an intensity of 250 mW/cm^2^. The experiment was repeated using silver nitrate instead of gold.

### 3.9. Transmission Electron Microscopy (TEM)

The morphology and particle size of the metal nanoparticles are examined via HR-TEM (JEOL, JEM-2100, Tokyo, Japan).

## 4. Conclusions

Photoinitiators A and B have proven to be effective for the synthesis of gold (AuNPs) and silver nanoparticles (AgNPs). The reactions can be completed rapidly within a few minutes under LED exposure. Our study suggests that the initiation mechanism of photoinitiator A involves intermolecular hydrogen abstraction followed by decarboxylation. Notably, ketyl radicals exhibited superior performance as reducing agents for HAuCl_4_ and AgNO_3_ compared to other radicals. The highly negative ∆G values (−3.514 eV for AuNPs and −3.459 eV for AgNPs) indicate the favorable nature of the production process using photoinitiator A.

On the other hand, the formation mechanism of nanoparticles in the photoprocess using photoinitiator B can be attributed to nucleation and growth processes. Photoinitiator B acts both as a reductant for Au^3+^ under irradiation and a surface stabilizer for the formed AuNPs. It is worth mentioning that photoinitiator B forms complexes with gold and silver at pH 6 and 12, as evidenced by the appearance of absorption bands at 298 nm and 325 nm, respectively.

The findings show that the concentration of AgNO_3_ had a significant impact on the size of silver nanoparticles. The diameters of the particles varied between 1 and 5 nm at 1 wt% and 3 wt% concentrations, whereas a rise in concentration to 5 wt% resulted in an increase in the diameter of silver nanoparticles to 16 nm. The average diameters of gold nanoparticles synthesized using both photoinitiators at different concentrations ranged between 1 and 4 nm. The results indicate that variations in the concentration of HAuCl_4_ have negligible effects on the size of gold nanoparticles in both photoinitiators. Additionally, at high concentrations of metal salts, aggregation was observed for both AgNPs and AuNPs.

## Data Availability

The data used in this study are available in the manuscript and Appendix A. For additional inquiries, please contact the corresponding author directly via email.

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
