# Peer review of "Photochemical Synthesis of Noble Metal Nanoparticles: Influence of Metal Salt Concentration on Size and Distribution"

_ijms, 2023, doi:10.3390/ijms241814018_

Round 1

Reviewer 1 Report

The work focuses on, firstly, the study of a photochemical reduction reaction using a single-component system of photoinitiators and, secondly, the use of this reaction for the synthesis of gold and silver nanoparticles. The reaction of reduction of silver ions to silver under irradiation has been known since ancient times and has been used in photogoraphy. The reduction reaction for gold is less studied. The authors successfully solved the set goals for the synthesis of the corresponding nanoparticles and found patterns in obtaining certain sizes of nanoparticles from the concentration of the initial substrates. The notes to the work (even taking into account the comment (9) are minor and nevertheless they must be corrected before the manuscript is accepted.

1.     Page 3. What is UVA (second paragraph from the bottom)? “..When the surfactant acted as a hydrogen donor, the growth of nanoparticles was significantly faster (CTAC)…” So what is CTAC (ibid.)? The introduction is full of many different abbreviations that do not always correspond to the text or there is no decoding in the text and are repeated many times. Or that's what it's for “...for the efficient synthesis of silver (Ag) and gold (Au)…”?? It is recommended to provide a single table of all abbreviations used and place it in the Introduction or Appendix.

2.     The experimental section is described very sparingly, poor and requires expansion and clarification. There is no indication on which instrument the fluorescence spectra were measured (for FigureS 4).

3.     Section 2.1. There is no indication of what type of photo initiators are given (despite a rather detailed explanation in the Introduction).

4.     Section 2.2. and Page 5 (last paragraph). To clarify what was the intensity of irradiation (is it milli-Watts or micro-Watts)?

5.     Section 2.3. Electrochemical equipment is not specified. Section 2.5. Clarify and correct: “…Shimadzo UV-1800 spectrophotometer (Shimadzo, Duisburg, Germany)…”. And not Kyoto, Japan?

6.     Section 3.2. Page 7 (the first paragraph from the bottom)ÑŽ “°OC ones”. What does it menace?

7.     Why do Sections 3.2 and 3.3 use different solvents: methanol and acetonitrile?

8.     In Section 3.3. Page 10. What is gold(III) chloride (AuB3). What precursor was used in the work "gold(III) chloride" or "(HAuCl4)"?
In Section 3.4.1. Page 8. "gold III chloride (HAuCl4)."
In Section 2.1. , Page 4. Gold (III) chloride hydrate (HAuCl4 99.99%).
There is no uniformity in the labelling of substances. It's confusing.

9.     This is a serious remark. Section 3.3. is written incomprehensibly, which brings confusion and lack of understanding. Figure S3, since electrochemical measurements play an important role in the manuscript, should be transferred to the main text. But it hardly needs to be done now: we understand that it is not always easy to get clear curves in ionic liquids, but the presented curves are simply not suitable. In addition, the authors have also marked the curves with a dotted line and it has become quite difficult to find the peaks. Secondly, the curves have oxidation peaks at 0.8V in the first figure and 0.8V , 1.5V in the second. This does not correspond at all to the data given in Table 1. How were the values obtained from Table 1? Clarification is required. According to the processes described in Section 3.3, it would be better to give a scheme (such as convincing schemes are given in the text of other sections). In the caption to Table 1, it is necessary to add a mention that the data were obtained on the basis of analysis for Figures S3 and S4. In the caption to Figure S3, specify the working electrode and the reference electrode. A description of the electrochemical installation, including all electrodes, should be given in the corresponding Experimental Section.

10.                       Figures 5 and 6. Too many details per figure. The font on the inserts in (a), (b), (c) is practically indistinguishable. It is recommended to make several figures. Some of the information can be moved to the SI.

11.                       Figures 8 and 9. You should increase the font size.

References

12.                       The list of 9 references out of 66 for the last five years (if include 2018). Almost everywhere there is no DOI index.

13.                       You need to use more modern references, e.g., In book: Catalytic Application of Nano-Gold Catalysts, Chapter 3, August 2016, DOI: 10.5772/64394; Carabineiro SAC (2019) Review. Supported Gold Nanoparticles as Catalysts for the Oxidation of Alcohols and Alkanes. Front. Chem. 7:702. doi: 10.3389/fchem.2019.00702.

14.                       Check the name of the journal to reference 31. Check the authors  (especially the author's names "Silver, E.; Nanoparticles, G.;.." are very well suited to the subject of the manuscript) and the reference 58 in general.

15.                       Part 6 does not contain any meaning, it should be removed.

Author Response

First of all, we appreciate the comments and suggestions by the reviewers, they are indeed valuable any by addressing them, the overall quality of the presented work will be significantly improved.

Point 1: Page 3. What is UVA (second paragraph from the bottom)? “..When the surfactant acted as a hydrogen donor, the growth of nanoparticles was significantly faster (CTAC)…” So what is CTAC (ibid.)? The introduction is full of many different abbreviations that do not always correspond to the text or there is no decoding in the text and are repeated many times. Or that's what it's for “...for the efficient synthesis of silver (Ag) and gold (Au)…”?? It is recommended to provide a single table of all abbreviations used and place it in the Introduction or Appendix.

Answer: We have revised the content by removing unclear sections and adding new references instead. Additionally, we have minimized abbreviations in the introduction to enhance readability for the reader.

Point 2: The experimental section is described very sparingly, poor and requires expansion and clarification. There is no indication on which instrument the fluorescence spectra were measured (for Figure S 4).

Answer: We have made the necessary changes in the experimental section (3.2) as per your instructions. Additionally, we have included the name of the instrument used for fluorescence measurement in section (3.6), specifically in line 488.

Point 3: Section 2.1. There is no indication of what type of photo initiators are given (despite a rather detailed explanation in the Introduction).

Answer: Compound A is a derivative of benzophenone and is classified as a type II photoinitiator. On the other hand, the identification of the new compound B's type is challenging due to unclear in terms of the yield and the nature of the radical center.

Point 4: Section 2.2. and Page 5 (last paragraph). To clarify what was the intensity of irradiation (is it milli-Watts or micro-Watts)?

Answer: It is milli-Watts

Point 5: Section 2.3. Electrochemical equipment is not specified. Section 2.5. Clarify and correct: “…Shimadzo UV-1800 spectrophotometer (Shimadzo, Duisburg, Germany)”. And not Kyoto, Japan?

Answer: It is a Shimadzu UV-1800 spectrophotometer (Shimadzu, Duisburg, Germany). The typo has been corrected on the new version of the manuscript (Lines 465-466)

Point 6: Section 3.2. Page 7 (the first paragraph from the bottom)ÑŽ “°OC ones”. What does it menace?

Answer:  It's a 0°C.

Point 7: Why do Sections 3.2 and 3.3 use different solvents: methanol and acetonitrile?

Answer: Acetonitrile's favorable attributes, including its minimal background current and broad potential range, make it suitable for examining reversible and irreversible redox reactions without significant disruption. Additionally, its low viscosity facilitates the smooth movement of reactants to the electrode surface, crucial for precise measurements. Furthermore, the chemical stability of acetonitrile within the standard potential range mitigates the risk of unwanted side reactions during cyclic voltammetry experiments.

Point 8: In Section 3.3. Page 10. What is gold(III) chloride (AuB3). What precursor was used in the work "gold(III) chloride" or "(HAuCl4)"?
In Section 3.4.1. Page 8. "gold III chloride (HAuCl4)."
In Section 2.1. , Page 4. Gold (III) chloride hydrate (HAuCl4 99.99%).
There is no uniformity in the labelling of substances. It's confusing.

Answer:   It is Gold (III) chloride hydrate (HAuCl4 99.99%). The mentioned positions have been modified and replaced with HAuCl4.

Point 9: This is a serious remark. Section 3.3. is written incomprehensibly, which brings confusion and lack of understanding. Figure S3, since electrochemical measurements play an important role in the manuscript, should be transferred to the main text.

Answer: Thank you for bringing this important matter to my attention. We have transferred Figure S3 to the main text and it is now referenced as Figure 5 (Line 253).

Point 10: It hardly needs to be done now: we understand that it is not always easy to get clear curves in ionic liquids, but the presented curves are simply not suitable. In addition, the authors have also marked the curves with a dotted line and it has become quite difficult to find the peaks.

Answer:  As depicted in Figure 5, there has been a change in the line style of the curve. It is important to emphasize that the specific point of interest in our calculations is Eox. This point is clearly visible in Figure 5. Furthermore, Figure 5b illustrates the oxidation peaks of compound B, and the peak with the lowest value (0.91 V) was utilized in the calculations.

Point 11: Secondly, the curves have oxidation peaks at 0.8V in the first figure and 0.8V , 1.5V in the second. This does not correspond at all to the data given in Table 1. How were the values obtained from Table 1? Clarification is required.

Answer: We have corrected the values and provided clarification in Figure 5. Additionally, we have recalculated and corrected the data in Table 1.

Point 12: According to the processes described in Section 3.3, it would be better to give a scheme (such as convincing schemes are given in the text of other sections). In the caption to Table 1, it is necessary to add a mention that the data were obtained on the basis of analysis for Figures S3 and S4.

Answer: Scheme 2 illustrates the potential formation of nanoparticles (NPs).

Point 13: In the caption to Figure S3, specify the working electrode and the reference electrode. A description of the electrochemical installation, including all electrodes, should be given in the corresponding Experimental Section.

Answer: We have added additional information in the experimental section (3.5) starting from line 474.

Point 14: Figures 5 and 6. Too many details per figure. The font on the inserts in (a), (b), (c) is practically indistinguishable. It is recommended to make several figures. Some of the information can be moved to the SI.

Point 15: Figures 8 and 9. You should increase the font size.

Answer: We have successfully moved some of the figures to the supporting information, as you suggested. Additionally, we have increased the size of the figures to ensure better visibility and clarity.

Point 15: References

  1. i) The list of 9 references out of 66 for the last five years (if include 2018). Almost everywhere there is no DOI index.
  2. ii) You need to use more modern references, e.g., In book: Catalytic Application of Nano-Gold Catalysts, Chapter 3, August 2016, DOI: 10.5772/64394; Carabineiro SAC (2019Review. Supported Gold Nanoparticles as Catalysts for the Oxidation of Alcohols and Alkanes. Front. Chem. 7:702. doi: 10.3389/fchem.2019.00702.

iii) Check the name of the journal to reference 31. Check the authors  (especially the author's names "Silver, E.; Nanoparticles, G.;.." are very well suited to the subject of the manuscript) and the reference 58 in general.

  1. iv) Part 6 does not contain any meaning; it should be removed.

Answer: We will address these issues in the final revision of our manuscript, and we appreciate the reviewer's careful reading and constructive criticism. Thank you again for your valuable feedback, which will undoubtedly improve the quality of our manuscript.

Reviewer 2 Report

This article by Aldebasi et al. describes the use of two different organic photoinitiators (PI) for the photoreduction of metal salts (Au or Ag) into nanoparticles. The articfle is difficult to read with a lot of typos and errors. The results are poorly presented (figures 5, 6, 8, and 9 are hardly readable) and the conclusions are superficial and vague. As it stands, I don’t believe this article reaches the standards of publication in IJMS and recommend rejection.

Some random concerns:

The unit of light intensity is wrong on page 4:  “250 m/cm²” should be “250 mW/cm²”

Page 5 “N-tert-butyl- -phenylnitrone (PBN) » should probably be « N-tert-butyl-alpha-phenylnitrone (PBN) »

Page 5 “bathochromic shift of 20 nm and 6 nm, respectively” should be “bathochromic shift of 20 nm and hypsochromic shift of 6 nm, respectively”. Or, more likely, the values of the peak maxima have been inverted between the reference and the compound (324 vs 330 nm).

The experimental section is too vague. There is not enough detail.

The write-up is very poor in some sections (3.4.1 for instance).

How can the authors explain that when more gold is added, the particles are smaller? Since the PI is also used as stabilizer, adding more gold vs PI should mean forming bigger NPs. But then again, this section is so poorly written, maybe I just did not understand what the authors meant…

How many NPs were counted on TEM images for the size distribution? Based on the y-axis of the graphs, it seems that less than 50 NPs were measured, which is far too few.

The notation Au3+Cl4- is just wrong and should be avoided in favor of [AuCl4]-

Scheme 4 is a total mess with notations like 3Au+Cl2- and 3Au+Cl4- that are not correct.

How is the rate of conversion increased when the concentration of gold is increased? Increasing gold vs PI is the equivalent of decreasing the wt% of catalyst and should induce slower conversion.

Have this MS edited by a native speaker

Author Response

We would like to express our gratitude for the insightful comments and suggestions provided by the reviewers. These inputs hold great value, and by incorporating them, we are confident that we can substantially enhance the overall quality of the work we have presented.

Point 1: The unit of light intensity is wrong on page 4: “250 m/cm²” should be “250 mW/cm²”

Answer: Thank you for bringing that to our attention. That has been corrected in line 194.

Point 2: Page 5 “N-tert-butyl- -phenylnitrone (PBN) » should probably be « N-tert-butyl-alpha-phenylnitrone (PBN) »

Answer:  The typo has been corrected in line 470.

Point 3: Page 5 “bathochromic shift of 20 nm and 6 nm, respectively” should be “bathochromic shift of 20 nm and hypsochromic shift of 6 nm, respectively”. Or, more likely, the values of the peak maxima have been inverted between the reference and the compound (324 vs 330 nm).

Answer: We have corrected that in line 164. Thank you for your observation.

Point 4: The experimental section is too vague. There is not enough detail.

Answer: We have made the necessary changes in the experimental section as per your instructions.

Point 5: The write-up is very poor in some sections (3.4.1 for instance).

Answer: Thank you for notifying me. We have made the necessary modification in Section 2.4.1, starting from line 265.

Point 6: How can the authors explain that when more gold is added, the particles are smaller? Since the PI is also used as stabilizer, adding more gold vs PI should mean forming bigger NPs. But then again, this section is so poorly written, maybe I just did not understand what the authors meant…

Answer: This is an experimental effect observed in our study and can be interpreted by the fact that: the gold precursor used is AuHCl4. We know that chlorine is a stripper and may be the particles synthesized in a medium containing so much precursor; undergo stripping which reduces their size.

Point 7: How many NPs were counted on TEM images for the size distribution? Based on the y-axis of the graphs, it seems that less than 50 NPs were measured, which is far too few.

Answer: We have treated the image provided by transmission electron microscope.

because nanoparticles are observed by TEM and necessarily on a much smaller scale than SEM. this is why we focus on a small number of particles, given the magnification used. the size distribution analysis is carried out using the well-known free software, used by most macroscopics in physics, chemistry, biology and medicine. this software is the ImageJ

ImageJ is a cross-platform, free and open source image processing and analysis software developed by the National Institutes of Health. It is written in Java and allows functionality to be added via plugins and macros.

In image analysis, ImageJ makes it possible to count particles, to evaluate their aspect ratios, to measure various quantities (distances, surfaces), to extract contour coordinates, etc.

Point 8: The notation Au3+Cl4- is just wrong and should be avoided in favor of [AuCl4]- Scheme 4 is a total mess with notations like 3Au+Cl2- and 3Au+Cl4- that are not correct.

Answer: Thank you for your feedback, which led to the correction of the scheme. It is worth mentioning that the proposed mechanism was inspired by the following reference (https://pubs.acs.org/doi/abs/10.1021/acs.langmuir.7b03192).

Point 9: How is the rate of conversion increased when the concentration of gold is increased? Increasing gold vs PI is the equivalent of decreasing the wt% of catalyst and should induce slower conversion.

Answer: The experimental findings indicate that when chloride is employed as a ligand, it has the potential to reduce particle size by means of etching and subdividing the particle into smaller entities. Consequently, this phenomenon results in an escalation in the particle count as the gold precursor content increases.

Round 2

Reviewer 2 Report

The article has been improved. I'm still not convinced that it meets the quality criteria for publication IJMS, I'll leave the final decision to the editor.

The article is understandable overall.